# Adjuvant Activity of Synthetic Lipid A of *Alcaligenes*, a Gut-Associated Lymphoid Tissue-Resident Commensal Bacterium, to Augment Antigen-Specific IgG and Th17 Responses in Systemic Vaccine

**DOI:** 10.3390/vaccines8030395

**Published:** 2020-07-20

**Authors:** Yunru Wang, Koji Hosomi, Atsushi Shimoyama, Ken Yoshii, Haruki Yamaura, Takahiro Nagatake, Tomomi Nishino, Hiroshi Kiyono, Koichi Fukase, Jun Kunisawa

**Affiliations:** 1Laboratory of Vaccine Materials, Center for Vaccine and Adjuvant Research, and Laboratory of Gut Environmental System, National Institutes of Biomedical Innovation, Health and Nutrition (NIBIOHN), Osaka 567-0085, Japan; wan@nibiohn.go.jp (Y.W.); hosomi@nibiohn.go.jp (K.H.); k-yoshii@nibiohn.go.jp (K.Y.); nagatake@nibiohn.go.jp (T.N.); pipipi_hiyoko2@yahoo.co.jp (T.N.); 2Graduate School of Pharmaceutical Sciences, Osaka University, Suita, Osaka 565-0871, Japan; 3Department of Chemistry, Graduate School of Science, Osaka University, Osaka 560-0043, Japan; ashimo@chem.sci.osaka-u.ac.jp (A.S.); yamaurah18@chem.sci.osaka-u.ac.jp (H.Y.); koichi@chem.sci.osaka-u.ac.jp (K.F.); 4Graduate School of Medicine, Osaka University, Osaka 565-0871, Japan; 5International Research and Development Center for Mucosal Vaccines, The Institute of Medical Science, The University of Tokyo, Tokyo 108-8639, Japan; kiyono@ims.u-tokyo.ac.jp; 6IMSUT Distinguished Professor Unit, The Institute of Medical Science, The University of Tokyo, Tokyo 108-8639, Japan; 7Graduate School of Medicine, Chiba University, Chiba 260-8670, Japan; 8Department of Medicine, School of Medicine and CU-UCSD Center for Mucosal Immunology, Allergy and Vaccine, University of California, Oakland, CA 92093-0063, USA; 9Graduate School of Dentistry, Osaka University, Osaka 565-0871, Japan; 10Department of Microbiology and Immunology, Graduate School of Medicine, Kobe University, Hyogo 650-0017, Japan; 11Research Organization for Nano & Life Innovation, Waseda University, Tokyo 162-0041, Japan

**Keywords:** *Alcaligenes faecalis*, Th17, lipid A

## Abstract

*Alcaligenes* spp. are identified as commensal bacteria and have been found to inhabit Peyer’s patches in the gut. We previously reported that *Alcaligenes*-derived lipopolysaccharides (LPS) exerted adjuvant activity in systemic vaccination, without excessive inflammation. Lipid A is one of the components responsible for the biological effect of LPS and has previously been applied as an adjuvant. Here, we examined the adjuvant activity and safety of chemically synthesized *Alcaligenes* lipid A. We found that levels of OVA-specific serum IgG antibodies increased in mice that were subcutaneously immunized with ovalbumin (OVA) plus *Alcaligenes* lipid A relative to those that were immunized with OVA alone. In addition, *Alcaligenes* lipid A promoted antigen-specific T helper 17 (Th17) responses in the spleen; upregulated the expression of MHC class II, CD40, CD80, and CD86 on bone marrow-derived dendritic cells (BMDCs); enhanced the production of Th17-inducing cytokines IL-6 and IL-23 from BMDCs. Stimulation with *Alcaligenes* lipid A also induced the production of IL-6 and IL-1β in human peripheral blood mononuclear cells. Moreover, *Alcaligenes* lipid A caused minor side effects, such as lymphopenia and thrombocytopenia. These findings suggest that *Alcaligenes* lipid A is a safe and effective Th17-type adjuvant by directly stimulating dendritic cells in systemic vaccination.

## 1. Introduction

The gut microbiota substantially affect the host immune system, such as the development and maturation of lymphoid tissues, promotion of intestinal IgA production, and recruitment of T cells and dendritic cells (DCs) [1,2,3,4,5]. For example, germ-free mice have smaller sized Peyer’s patches (PPs) and fewer IgA-producing plasma cells, CD4^+^ T cells, and DCs in the intestine [6,7]. Technological advances in the analysis of commensal bacteria have enabled the identification of specific functions of bacteria involved in the regulation of host immunity. For example, *Bacteroides* and *E. coli* induce the formation of isolated lymphoid follicles in the intestine [8]. Segmented filamentous bacteria induce the differentiation of T helper 17 (Th17) cells and IgA production [9], whereas *Clostridium* strains induce the accumulation of regulatory T cells in the intestine [10].

Although many studies have focused on commensal bacteria that inhabit the intestinal lumen or epithelium, we previously reported that commensal bacteria are also resident inside gut-associated lymphoid tissue (GALT), such as PPs [11]. The predominant resident symbiotic bacteria in PPs are *Alcaligenes* spp., which are Gram-negative facultatively anaerobic bacteria and belonged to the class of the Gram-negative Betaproteobacteria. Our previous study revealed that *Alcaligenes* spp. That enter the PPs via microfold cells (M cells) are then captured by DCs and induce the production of IL-6 and IL-10, leading to the subsequent secretion of IgA in the intestine [12].

Bacterial components, such as polysaccharide and lipopolysaccharide (LPS), are well known to show activating effects on host immune responses [13,14]; therefore, they have been considered as vaccine adjuvants [15,16]. Adjuvants enhance the efficacy of vaccines, which can reduce the amount of antigen required, thereby lowering cost and improving efficacy in populations of poor responders, such as neonates, immunocompromised individuals, and the elderly [17,18]. In this regard, our previous study showed that *Alcaligenes*-derived LPS is a weak agonist for toll-like receptor 4 (TLR4), promoting the production of IgA-enhancing cytokines, such as IL-6, from DCs [19]. When used as an adjuvant, *Alcaligenes*-derived LPS consistently enhances antigen-specific immune responses, including antibody production and Th17 response, without excessive inflammation [19]. LPS consists of a core, an O-antigen, and lipid A [20]. The chemical structure of lipid A varies among bacteria and shows different biological activities [20,21]. Based on our previous findings on the adjuvant activity of *Alcaligenes* LPS, we aimed to examine the efficacy and safety of chemically synthesized *Alcaligenes* lipid A when used as a vaccine adjuvant.

## 2. Materials and Methods

### 2.1. Mice

Female BALB/c mice (aged 8 weeks) were purchased from CLEA Japan, Inc. (Tokyo, Japan) and kept in a specific pathogen-free environment at the National Institutes of Biomedical Innovation, Health and Nutrition (NIBIOHN) (Osaka, Japan). All animal experiments were conducted in accordance with the Animal Care and Use Committee guidelines of the NIBIOHN (Approval Nos. DS25-2, DS25-3).

### 2.2. Preparation of Lipid A

*Alcaligenes* lipid A was chemically synthesized as previously described [22]. *Alcaligenes* lipid A was dissolved in dimethyl sulfoxide (DMSO) (Nacalai Tesque, Inc., Kyoto, Japan) at a concentration of 1 mg/mL for the stock. For immunization, 1 mg/mL of *Alcaligenes* lipid A was diluted to the concentration of 10 μg/mL with PBS to mix with OVA in PBS.

### 2.3. Immunization

On days 1 and 10, mice were injected subcutaneously in the back with a total volume of 0.2 mL vaccine that contained with 1 μg of OVA (Sigma-Aldrich, St. Louis, MO, USA) alone, 10 μg of OVA alone, or 1 μg of OVA plus 1 μg of *Alcaligenes* lipid A.

### 2.4. Enzyme-Linked Immunosorbent Assay

Ninety-six-well immunoplates (Thermo Fisher Scientific, Inc., Waltham, MA, USA) were coated with 1 mg/mL OVA diluted in PBS and left overnight at 4 °C. The solution in the plates was removed, and 1% (*w*/*v*) BSA (Nacalai Tesque, Inc.), dissolved in PBS, was added to the plates and incubated at room temperature for 2 h. Plates were washed 3 times with wash buffer (0.05% [*v*/*v*] Tween 20 [Nacalai Tesque, Inc.] in PBS). Serum samples, diluted with 1% (*w*/*v*) BSA and 0.05% (*v*/*v*) Tween 20 in PBS, were plated in serial dilutions and then, incubated for 2 h at room temperature. Goat anti-mouse IgG, IgG1, IgG2b, and IgG3, conjugated with horseradish peroxidase (SouthernBiotech, Inc., Birmingham, AL, USA), were diluted at a dilution of 1:4000 with 1% (*w*/*v*) BSA (Nacalai Tesque, Inc.) and 0.05% (*v*/*v*) Tween 20 in PBS, and then, incubated for 1 h at room temperature. Plates were washed 3 times with wash buffer. Tetramethylbenzidine peroxidase substrate (SeraCare Life Sciences, Inc., Milford, MA, USA) was added to the plates and then, incubated for 2 min at room temperature. Next, 0.5 N HCl (Nacalai Tesque, Inc.) was added to the plates. The absorbance of serum samples was measured at 450 nm with an iMark™ Microplate Absorbance Reader (Bio-Rad Laboratories, Inc., Hercules, CA, USA).

### 2.5. T Cell Assay

One week after the last immunization, single cell suspensions were prepared from the spleen of mice with a 100 μm filter (Thermo Fisher Scientific, Inc.) and treated with 1 mL of red blood cell lysis buffer (1.5 M NH_4_Cl (Nacalai Tesque, Inc.), 100 mM KHCO_3_ (Nacalai Tesque, Inc.), and 10 mM EDTA-2Na (Nacalai Tesque, Inc.)) for 1 min at room temperature. Splenic CD4^+^ T cells were isolated with a magnetic cell separation system with CD4 (L3T4) microbeads (Miltenyi Biotec, Bergisch Gladbach, Germany) and MS Columns (Miltenyi Biotec), following the manufacturer’s protocol. Antigen presenting cells (APCs) were prepared from the single cell suspensions and under 30 Gy of ionizing radiation by MBR-1520R-4 (Hitachi, Ltd., Tokyo, Japan). Splenic CD4^+^ T cells (2 × 10^5^ cells/well) were seeded into Nunc™ 96-Well, Nunclon Delta-Treated, U-Shaped-Bottom Microplate (Thermo Fisher Scientific, Inc.) and incubated for 4 days with APCs (1 × 10^4^ cells/well) in 1 mg/mL OVA dissolved in complete RPMI-1640 medium (RPMI-1640 medium supplemented with 10% fetal bovine serum (Life Technologies, Thermo Fisher Scientific, Inc., Waltham, MA, USA), 1% 100 mM sodium pyruvate solution (100×) (Nacalai Tesque, Inc.), 1% penicillin-streptomycin mixed solution (Nacalai Tesque, Inc.), and 0.1% 2-Mercaptoethanol (Gibco, Thermo Fisher Scientific, Inc., Waltham, MA, USA)). Cell viability was measured with CyQUANT^®^ Cell Proliferation Assay Kits (Invitrogen, Thermo Fisher Scientific, Inc., Waltham, MA, USA). The absorbance of the cells was measured at 485/535 nm with an ARVO X2 (PerkinElmer, Yokohama, Japan) fluorescence microplate reader. The cell supernatants were collected and the cytokines interferon γ (IFN-γ), IL-4, and IL-17 were detected with a BD™ Cytometric Bead Array (CBA) Mouse Th1/Th2/Th17 Cytokine Kit (BD Biosciences, San Jose, CA, USA), following the manufacturer’s protocol, and analyzed with a MACSQuant^®^ Analyzer (Miltenyi Biotec).

### 2.6. Preparation of Bone Marrow-Derived Dendritic Cells and Splenic Dendritic Cells

Bone marrow-derived dendritic cells (BMDCs) were cultured as previously described [19]. Cells were harvested from the femoral bone marrow (BM) of female BALB/c mice aged 4–5 weeks. The BM cells were treated with red blood cell lysis buffer for 5 min, and then, washed in complete RPMI-1640 medium. BMDCs were cultured in complete RPMI-1640 medium containing 20 ng/mL GM-CSF (PeproTech, Rocky Hill, NJ, USA). Half of the culture medium was replaced with fresh medium every 2 days. On day 6, BMDCs were purified using the MACS magnetic cell separation system with CD11c microbeads (Miltenyi Biotec) and LS Columns (Miltenyi Biotec), following the manufacturer’s protocol. For analyzing splenic dendritic cells, single cell suspensions were prepared from the spleen of mice that were subcutaneously injected once after 4 h from the injection, with a 100 μm filter (Thermo Fisher Scientific, Inc.) and treated with 1 mL of red blood cell lysis buffer (1.5 M NH_4_Cl (Nacalai Tesque, Inc.), 100 mM KHCO_3_ (Nacalai Tesque, Inc.), and 10 mM EDTA-2Na (Nacalai Tesque, Inc.)) for 1 min at room temperature.

### 2.7. Measurement of Cytokines

BMDCs (1 × 10^5^ cells/well) or human peripheral blood mononuclear cells (PBMCs) (2 × 10^5^ cells/well, FUJIFILM Wako Pure Chemical, Osaka, Japan) were seeded into a Nunc™ 96-Well, Nunclon Delta-Treated, U-Shaped-Bottom Microplate (Thermo Fisher Scientific, Inc.) and incubated with various concentrations (0.1 or 1 ng/mL) of *Alcaligenes* lipid A for 24 (PBMCs) or 48 (BMDCs) h at 37 °C. The cell culture supernatants were collected for the detection of cytokines via BD™ CBA Mouse or Human Inflammation Kit (BD Biosciences), LEGEND MAX™ Human IL-1β ELISA Kit (BioLegend, San Diego, CA, USA), and LEGEND MAX™ Mouse IL-23 (p19/p40) ELISA Kit (BioLegend).

### 2.8. Flow Cytometry Analysis

BMDCs or splenic cells were incubated for 15 min at room temperature with 5 μg/mL anti-CD16/32 antibody (Fc Block; clone: 93) (BioLegend), to avoid non-specific staining, and 7-AAD Viability Staining Solution (BioLegend), to detect dead cells. BMDCs or splenic cells were then stained for 30 min at 4 °C with following antibodies. For BMDCs, FITC-anti-I-A^d^ (BioLegend; clone: 39-10-8), PE-anti-CD80 (BioLegend; clone: 16-10A1), APC-Cy7-anti-CD86 (BioLegend; clone: GL-1), and PE-Cy7-anti-CD40 (BioLegend; clone: 3/23) were used. For splenic cells, FITC-anti-I-A^d^ (BioLegend; clone: 39-10-8), PE-anti-CD80 (BioLegend; clone: 16-10A1), and Brilliant Violet 421-anti-CD11c (BioLegend; clone: N418) were used. Samples were analyzed with a MACSQuant^®^ Analyzer (Miltenyi Biotec). Flow cytometry data were analyzed with FlowJo, LLC Software 10.2 (BD Biosciences).

### 2.9. Measurement of Blood Cells and Body Temperature in Mice

Blood samples (100 μL) were mixed with 1.5μL of 10 mM EDTA-2Na (Nacalai Tesque, Inc.) and then, diluted 1:6 with saline solution (Otsuka Pharmaceutical Co., Ltd., Tokyo, Japan). The number of lymphocytes and platelets was measured with a Vet Scan HMII hematology analyzer (Abaxis, Union City, CA, USA). Body temperature was obtained by measuring the rectal temperature.

### 2.10. Statistical Analyses

The data are presented as mean ± SD. Statistical analyses were performed using Student’s *t*-test and one-way ANOVA with the Bonferroni post hoc test by PRISM (GraphPad Software, San Diego, CA, USA).

## 3. Results

### 3.1. Enhancement of OVA-Specific Antibody Responses by Alcaligenes Lipid A

To examine the effects of *Alcaligenes* lipid A on the induction of antigen-specific antibody responses, mice were subcutaneously injected with OVA alone or OVA plus *Alcaligenes* lipid A on days 1 and 10. One week after the last immunization, serum was collected to measure the levels of OVA-specific antibodies by enzyme-linked immunosorbent assay (ELISA). Levels of OVA-specific serum IgG were higher in mice immunized with 1 μg of OVA plus *Alcaligenes* lipid A than in mice immunized with 1 or 10 μg of OVA alone (Figure 1a). We further found that the OVA-specific IgG induced by immunization with 1 μg of OVA plus *Alcaligenes* lipid A contained higher levels of OVA-specific serum IgG1, IgG2b, and IgG3 (Figure 1b). These results indicate that *Alcaligenes* lipid A enhances OVA-specific IgG responses.

### 3.2. Enhancement of OVA-Specific T Cell Responses by Alcaligenes Lipid A

We next examined the effects of *Alcaligenes* lipid A on OVA-specific T cell responses. To examine their proliferation and cytokine production, splenic CD4^+^ T cells were collected from mice immunized with OVA alone or OVA plus *Alcaligenes* lipid A. Splenic CD4^+^ T cells isolated from mice that had been immunized with OVA plus *Alcaligenes* lipid A proliferated significantly more upon antigen re-stimulation than did those isolated from mice that had been immunized with OVA alone (Figure 2a), indicating that *Alcaligenes* lipid A promoted the proliferation of OVA-specific CD4^+^ T cells.

Splenic CD4^+^ T cells from mice that had been immunized with OVA plus *Alcaligenes* lipid A also produced significantly higher levels of IL-17 compared with mice that had been immunized with OVA alone, but there were no differences in the production of IFN-γ and IL-4 (Figure 2b). These results collectively indicate that *Alcaligenes* lipid A preferentially induces Th17 responses.

### 3.3. Activation of Bone Marrow-Derived Dendritic Cells And Splenic Dendritic Cells by Alcaligenes Lipid A

To examine the effects of *Alcaligenes* lipid A on the activation of DCs, we measured the expression level of MHC class II and costimulatory molecules (CD80 and CD86, related to activation of T cells; CD40, related to antibody production from B cells) on BMDCs. Stimulation with *Alcaligenes* lipid A significantly increased the expression level of MHC class II, CD80, CD86, and CD40 on BDMCs in a dose-dependent manner (Figure 3a). Furthermore, consistent with the preferential induction of Th17 cells, *Alcaligenes* lipid A activated BMDCs to produce IL-6 and IL-23, which are cytokines involved in Th17 cell differentiation and subsequent stabilization (Figure 3b). In addition, we also analyzed the expression level of MHC class II and costimulatory molecules CD80 in vivo. Mice subcutaneously injected with OVA and *Alcaligenes* lipid A significantly increased the numbers of the CD11c^+^ MHC class II^+^ CD80^+^ DCs in spleen (Appendix A). These findings collectively indicate that *Alcaligenes* lipid A activates DCs and creates an environment that is preferential for the induction of Th17 cells.

### 3.4. Activation of Human Immune Cells by Alcaligenes Lipid A

To consider the effects of *Alcaligenes* lipid A on human immune cells, we measured the production of cytokines from human PBMCs from two individuals; both showed increased levels of IL-6 and IL-1β, which are required for the initiation and stabilization of human Th17 cells (Appendix A). These results indicate that *Alcaligenes* lipid A activate human immune cells, especially those that contribute toward the induction of Th17 cells.

### 3.5. Safety of Alcaligenes Lipid A

To investigate the safety of *Alcaligenes* lipid A, we examined the numbers of lymphocytes and platelets in the blood of mice 24 h after immunization. The numbers of lymphocytes and platelets from mice after immunization with OVA plus *Alcaligenes* lipid A were reduced compared with those from mice that were immunized with OVA alone; however, the levels remained within the normal physiological ranges of mice (Figure 4a).

We further surveyed body temperature at 0, 1, 8, and 24 h after immunization. Immunization with OVA and *Alcaligenes* lipid A or OVA alone led to no significant changes in body temperature (Figure 4b). Similarly, for both groups, body weight did not significantly change after immunization (Figure 4c). Collectively, these results indicate that immunization with *Alcaligenes* lipid A did not cause severe side effects in mice.

## 4. Discussion

In this study, we have demonstrated the efficacy and safety of *Alcaligenes* lipid A when used as an adjuvant in systemic vaccination. Lipid A is a core component of LPS, and its activity related to TLR4 agonist or antagonist is determined by its structure, including the number of acyl chains and phosphate groups, and the length of acyl chains [23,24]. For use as a vaccine adjuvant, the balance of the structure–activity relationship between efficacy and safety is the issue that has to be considered. For example, 3-*O*-desacyl-4’-monophosphoryl lipid A (MPL) is generated by modification of the dephosphorylating of one phosphate group of the lipid A of *Salmonella minnesota* to reduce its toxicity [25,26,27,28,29,30]. Similarly, the monophosphoryl lipid A from *Bordetella pertussis* and *Escherichia coli* has also been reported as an effective adjuvant [31,32,33,34,35]. *Alcaligenes* lipid A has the same structure as *E. coli*, with six acyl chains and two phosphate groups. However, the length of the acyl chains in the structure of *Alcaligenes* lipid A is shorter than that of *E. coli* [22], suggesting that the structural uniqueness of *Alcaligenes* lipid A provides effectiveness and safety when used as a vaccine adjuvant. Previous works on lipid A as a vaccine adjuvant focus on lipid A from *S. minnesota*, *B. pertussis*, and *E. coli*, which are pathogenic bacteria. Our studies showed that when lipid A from *Alcaligenes*, a symbiotic bacterium that resides inside the PPs of various mammalian species, is used as vaccine adjuvant in mice experiments, it is effective and safe without modifying its structure [22].

One of the unique properties of *Alcaligenes* lipid A is the induction of antigen-specific Th17 responses. In contrast, MPL from *S. minnesota* and monophosphoryl lipid A from *B. pertussis* and *E. coli* promote IFN-γ-producing Th1 responses in mice [30,31,34]. Th17 cells are important in conferring protection against extracellular bacterial and fungal infections; therefore, *Alcaligenes* lipid A could be useful as an adjuvant in vaccines against these types of infections.

To understand the mechanism underlying the preferential induction of Th17 responses, the effects of *Alcaligenes* lipid A on DCs were examined. DCs present antigen to T cells via MHC class II and costimulatory molecules, such as CD80 and CD86 [36,37], expression of which are upregulated by *Alcaligenes* lipid A. Moreover, the cytokine pattern produced by DCs determine the direction of T helper cell differentiation. For example, IFN-γ and IL-12 induce the differentiation of Th1 cells, whereas IL-4 induces the differentiation of Th2 cells. Transforming growth factor beta (TGF-β) and IL-6 promote the differentiation of Th17 cells, and IL-23 promotes the stabilization of Th17 cells in mice [38,39,40]. Consistent with our previous reports about *Alcaligenes*-mediated activation of DCs [11,12,19], *Alcaligenes* lipid A increased IL-6 and IL-23 production from murine BMDCs. Furthermore, human PBMCs that were stimulated by *Alcaligenes* lipid A had increased production of IL-6 and IL-1β, which contribute to the differentiation and stabilization of human Th17 cells [38,39,40,41]. IL-23 was also produced by the PBMCs; however, unlike IL-6 and IL-1β, the reactivity was different among the individuals. This difference between individuals indicates that *Alcaligenes* lipid A may have multiple pathways to induce human Th17 development [38,39,40,41].

In addition, *Alcaligenes* lipid A enhanced the expression of CD40 on BMDCs, suggesting that *Alcaligenes* lipid A could induce antibody production via the T cell independent pathway. CD40 expressed on DCs plays a role in regulating B cell proliferation by the direct interaction via CD40L expressed on B cells, leading to enhance IgG production [42]. Consistently, we found that *Alcaligenes* lipid A induced the production of antigen-specific IgG1, IgG2b, and IgG3 in the serum. Similarly, it was reported that MPL of *S. minnesota* enhances the production of IgG2a and IgG2b [26,28], and monophosphoryl lipid A of *B. pertussis* and *E. coli* both enhance the production of IgG2a [31,35]. In addition, MPL of *S. minnesota* and monophosphoryl lipid A of *B. pertussis* and *E. coli* slightly increase IgG1 production [26,28,31,35]. The results of the current study are consistent with previous works that demonstrated that Th1 cells produce IFN-γ to induce IgG2a and IgG2b, while Th2 cells produce IL-4 to induce IgG1 [43,44]. Th17 cells induce IgG1, IgG2b, and IgG3 via the secretion of IL-17A or IL-21 [45]. IgG1 and IgG3 contribute toward the activation of phagocytosis by macrophages and neutrophils and activation of complement, both of which help to defend against bacterial infections [46]. These findings indicate that *Alcaligenes* lipid A could be an effective adjuvant in vaccines against extracellular bacterial infections.

Adjuvant safety is another important factor for clinical use. For example, complete Freund’s adjuvant is a well-known strong adjuvant that is widely used in animal experiments; however, it cannot be used clinically because of side effects, such as severe inflammation. In this regard, *Alcaligenes* lipid A did not cause severe side effects, which is consistent with our previous studies on *Alcaligenes* LPS. Injection with 1 μg of *Alcaligenes* lipid A showed efficacy and without severe side effects such as lymphopenia and thrombocytopenia in mice. However, we noticed that injection with more than 1 μg of *Alcaligenes* lipid A caused lymphopenia and thrombocytopenia in mice; therefore, when considering clinical use, it is necessary to reduce the side effects of *Alcaligenes* lipid A by setting a safe dose or modifying the structure.

## 5. Conclusions

We have demonstrated the efficacy and safety of chemically synthesized *Alcaligenes* lipid A when used as an adjuvant in systemic vaccination. *Alcaligenes* lipid A promoted both antigen-specific IgG antibody and Th17 responses in mice by directly stimulating DCs. Stimulation with *Alcaligenes* lipid A also induced the production of IL-6 and IL-1β in human PBMCs, suggesting a potency to be applied for use in human.

## Figures and Tables

**Figure 1 vaccines-08-00395-f001:**
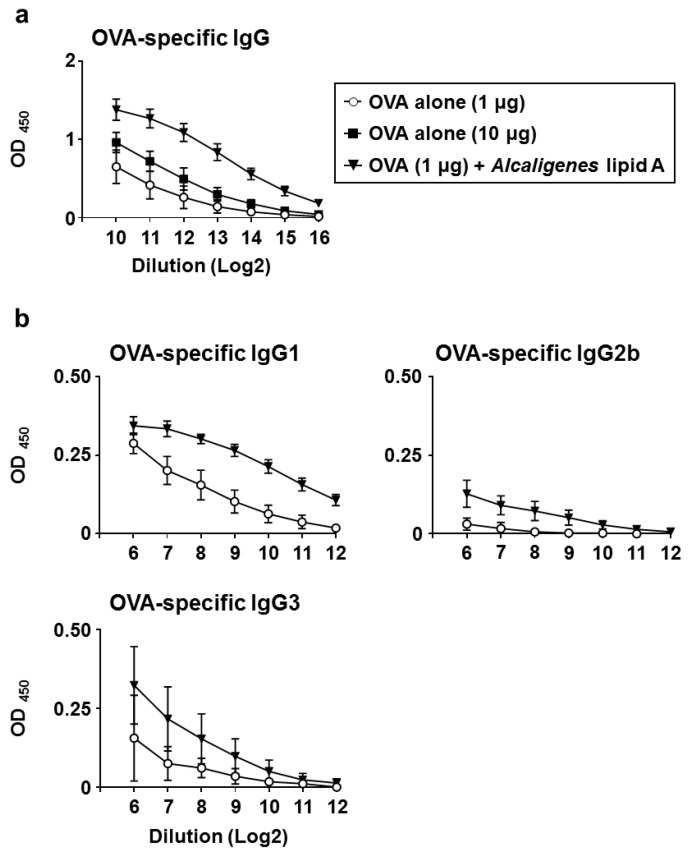
Enhancement of OVA-specific antibody responses by *Alcaligenes* lipid A. Mice were subcutaneously immunized with OVA plus *Alcaligenes* lipid A (1 μg) or OVA alone (1 or 10 μg) on days 1 and 10. One week after the last immunization, serum was collected to measure with ELISA to determine the levels of OVA-specific IgG (**a**), IgG1, IgG2b, and IgG3 (**b**) (*n* = 7 per group). Data are a combination of two independent experiments.

**Figure 2 vaccines-08-00395-f002:**
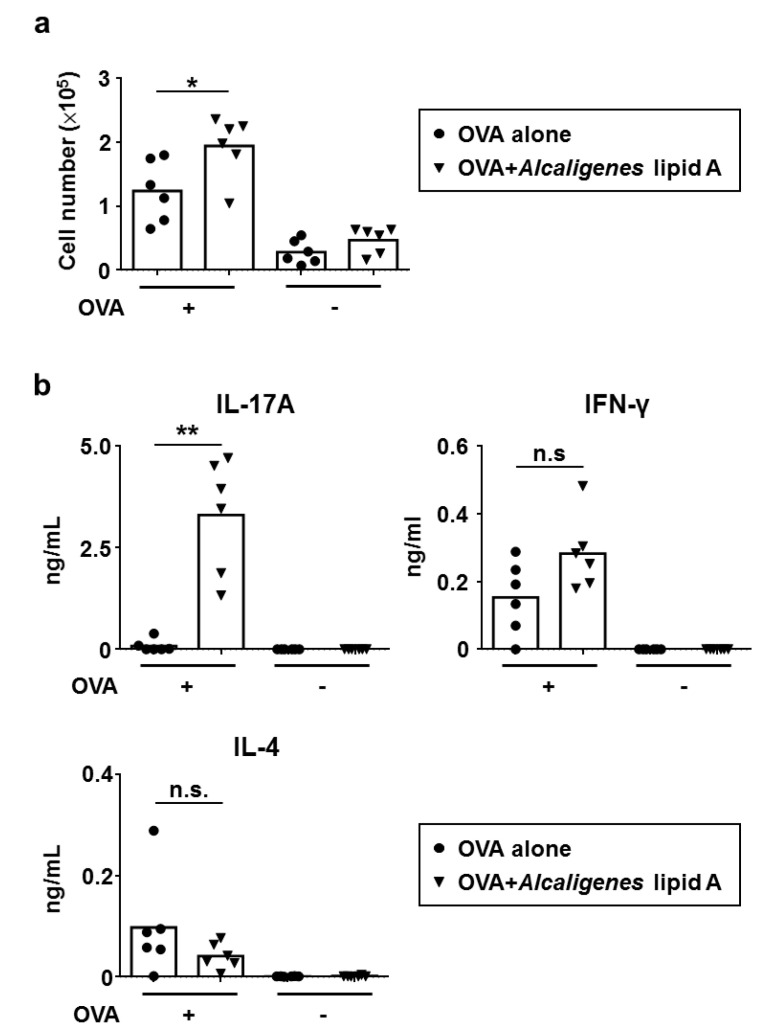
Enhancement of OVA-specific T cell responses by *Alcaligenes* lipid A. Splenic CD4^+^ T cells were collected from mice subcutaneously immunized with OVA alone (1 μg) or plus *Alcaligenes* lipid A (1 μg). After ex vivo stimulation with OVA (+) or not (-), the number of T cells (**a**) and the production of IL-17, IFN-γ, and IL-4 in the culture supernatants (**b**) were measured (*n* = 6 per group). Data are a combination of two independent experiments and analyzed by Student’s *t*-test (* *p* < 0.05; ** *p* < 0.01; n.s.: not significant).

**Figure 3 vaccines-08-00395-f003:**
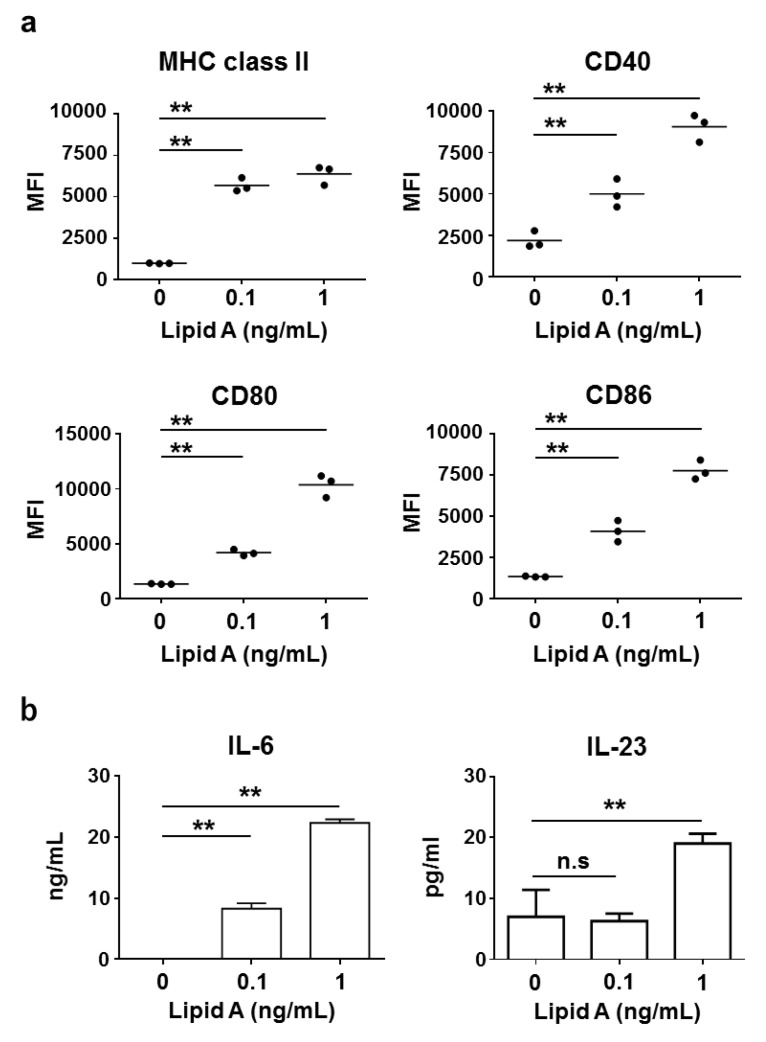
Activation of BMDCs by *Alcaligenes* lipid A. Murine BMDCs were stimulated with 0 (control), 0.1 or 1 ng/mL *Alcaligenes* lipid A. After incubation for 48 h, the expression of MHC class II, CD80, CD86, and CD40 (**a**) and the production of IL-6 and IL-23 in the culture supernatant (**b**) were analyzed (*n* = 3 per group). Data are representative of two independent experiments and analyzed by one-way ANOVA (** *p* < 0.01; n.s.: not significant).

**Figure 4 vaccines-08-00395-f004:**
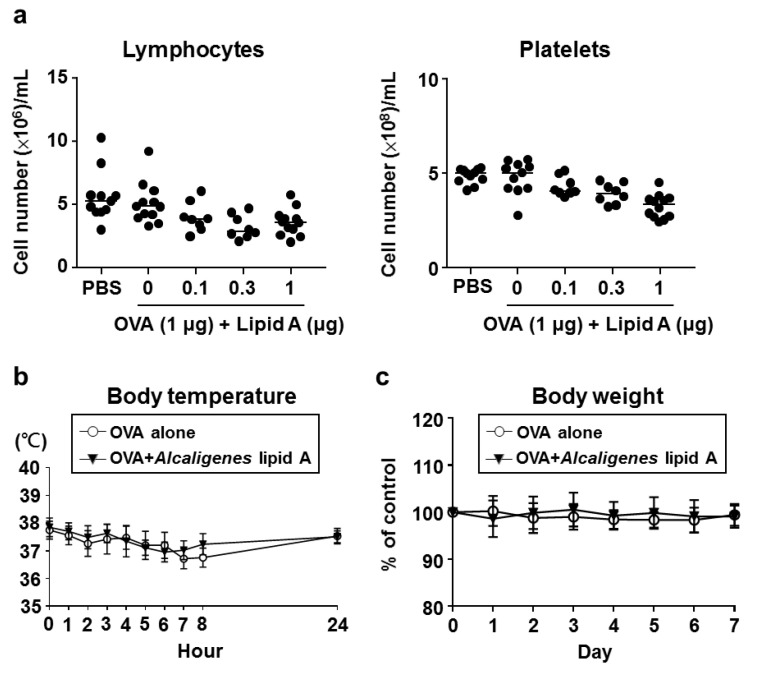
Safety of *Alcaligenes* lipid A. Mice were immunized with OVA or OVA plus *Alcaligenes* lipid A. The numbers of lymphocytes and platelets in the blood were measured 24 h after immunization (**a**). Body temperature (**b**) was monitored at 0, 1, 8, and 24 h; and body weight (**c**) for 7 days after immunization (*n* = 8–12 per group). Data are a combination of three independent experiments.

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
