# Peer review of "Adjuvant Activity of Synthetic Lipid A of Alcaligenes, a Gut-Associated Lymphoid Tissue-Resident Commensal Bacterium, to Augment Antigen-Specific IgG and Th17 Responses in Systemic Vaccine"

_vaccines, 2020, doi:10.3390/vaccines8030395_

Round 1

Reviewer 1 Report

Authors evaluate the adjuvant capacity of a chemically synthetized lipid A from Alcaligenes. The study aims to show that this adjuvant lipid enhances activation of dendritic cells and potentially induce Th17 cells in a model of OVA- sensitized mice. The effects of the adjuvant are of potential benefit for poor responders to bacterial and fungal infections. Two major concerns of the study are pointed:

Identification of the Th17 cells and DCs responding to the adjuvant stimulation was not provided. Results in Figure 1 and Figure 2b and part of Figure 3b are just repeated using the synthetic form of the adjuvant without further exploration. Flow cytometry determinations in tissues or ex vivo response of immunized mice would be more definitive that correlates of cytokines associated to Th17 cells and activation of DCs in supernatants. Why the proportion of Th17 cells and surface activation markers of DCs are not shown in immunized mice instead of supernatant correlates? The novelty of the findings is adversely impacted because most of the results seem to reproduce similar published effects of Alcaligenes LPS by the same group (Shibata et al., 2017. doi: 10.1038/mi.2017.103).

Proof that the synthetized lipid was not inducing excessive inflammatory response is missing. Their previous publication with Alcaligenes LPS is cited to support the effect of the synthetic lipid.  

Other minor points:

Data of OVA-specific IgA, is missing of the figure 1b.

Statistical significance of Figure 4 is missing. The figure lack of enough quality enough to be in the core of the manuscript and should be presented as supplementary figure. Otherwise, should increase the number of determinations to show reproducibility and statistical significance of results.

The dose of lipid that induce lymphopenia and the lack of immune suppression statements are not clear. Is lymphopenia absent in 1ug treated mice? Although the number of T cells in immunized mice (1ug) were increased during the ex vivo culture and OVA stimulation (Figure 2a) where the mice, source of cells, with any degree of lymphopenia? Which cells populations were impacted in lymphogenic mice? Is autoimmunity induced by the Th17 response associated to it?

Author Response

Authors evaluate the adjuvant capacity of a chemically synthetized lipid A from Alcaligenes. The study aims to show that this adjuvant lipid enhances activation of dendritic cells and potentially induce Th17 cells in a model of OVA- sensitized mice. The effects of the adjuvant are of potential benefit for poor responders to bacterial and fungal infections.

Major points:

Identification of the Th17 cells and DCs responding to the adjuvant stimulation was not provided. Results in Figure 1 and Figure 2b and part of Figure 3b are just repeated using the synthetic form of the adjuvant without further exploration. Flow cytometry determinations in tissues or ex vivo response of immunized mice would be more definitive that correlates of cytokines associated to Th17 cells and activation of DCs in supernatants. Why the proportion of Th17 cells and surface activation markers of DCs are not shown in immunized mice instead of supernatant correlates? The novelty of the findings is adversely impacted because most of the results seem to reproduce similar published effects of Alcaligenes LPS by the same group (Shibata et al., 2017. doi: 10.1038/mi.2017.103).

 We appreciate very much for the important suggestions. We agree that flow cytometry determinations in tissues or ex vivo response of immunized mice is more definitive, but it has limitation to detect the antigen-specific Th17 cells due to their low frequency. Therefore, we employed ex vivo OVA stimulation system as a standard protocol frequently used to analyze antigen-specific T cell responses. We also agree that it is more definitive to analyze DCs in tissues by flow cytometry analysis. In this issue, we examined DCs in mice receiving subcutaneous immunization with OVA with or without Alcaligenes lipid A. After 4 hours from the injection and found that the subcutaneous injection with OVA plus Alcaligenes lipid A increased numbers of MHC class II+ CD80+ activated CD11c+ DCs in the spleen (Supplementary Fig. 1). This result is mentioned at line 201 - 204 on page 5 of the revised manuscript.

Proof that the synthetized lipid was not inducing excessive inflammatory response is missing. Their previous publication with Alcaligenes LPS is cited to support the effect of the synthetic lipid. 

 Thanks for the helpful comment. Regarding this comment, we found that Alcaligenes lipid A induced serum TNFα elevation in a dose-dependent manner at 6 hr after injection but reached to basal levels at 24 hr (Appendix Fig. 1). Additionally, even maximum concentration of TNFα induced by Alcaligenes lipid A was much lower than that of injection with 5 mg/kg of E.coli lipopolysaccharides, a typical dose to induce massive inflammation (Ogino H, et al., J Infect Chemother. 2009; Somann JP, et al., PLoS One. 2019). Together with data presented in our original manuscript (Fig. 4), this finding indicates that Alcaligenes lipid A does not induce excessive inflammatory responses.

Minor points:

Data of OVA-specific IgA, is missing of the figure 1b.

 We apologize for this mistake and have corrected it.

Statistical significance of Figure 4 is missing. The figure lack of enough quality enough to be in the core of the manuscript and should be presented as supplementary figure. Otherwise, should increase the number of determinations to show reproducibility and statistical significance of results.

 According to the reviewer’s suggestion, we have presented it as supplementary figure 2 in the revised manuscript.

The dose of lipid that induce lymphopenia and the lack of immune suppression statements are not clear. Is lymphopenia absent in 1ug treated mice? Although the number of T cells in immunized mice (1ug) were increased during the ex vivo culture and OVA stimulation (Figure 2a) where the mice, source of cells, with any degree of lymphopenia? Which cells populations were impacted in lymphogenic mice? Is autoimmunity induced by the Th17 response associated to it?

 We apologize for the insufficient information. Accordingly, we have added the information at line 279 - 280 on page 6. We found that the numbers of lymphocytes and platelets remained within normal physiological range in mice receiving 1 μg of Alcaligenes lipid A. Consistently, we did not see any reduction of cell numbers in the spleen in our ex vivo OVA stimulation assay (Fig. 2a).

 Regarding cell populations impacted in lymphogenic mice, a previous study showed that both B and T lymphocytes are impacted in lymphopenia (Chandra R, et al., J Leukoc Biol. 2008) and another study indicated that promoted production of inflammatory cytokines such as TNFα by LPS injection induced apoptosis-mediated lymphopenia (Hattori Y, et al., J Pharmacol Sci. 2010; Aziz M, et al., Cell Death Dis. 2014). Thus, it is considered that autoimmunity induced by the Th17 response is not associated with lymphopenia.

Reviewer 2 Report

Adjuvant activity of synthetic lipid A of Alcaligenes, a gut-associated lymphoid tissue-resident commensal bacterium, to augment antigen-specific IgG and Th17 responses in systemic vaccine.

Y. Wang et al.

This is a well constructed and executed study and a well-crafted paper. The authors have determined that lipid A from Alcaligenes is an effective adjuvant when using OVA as an antigen. It may be a well suited Th17-type adjuvant for systemic vaccination due to stimulation of dendritic cells.

Line 88:  "… 1 mg/mL and then mixed with OVA that was diluted in PBS for immunization."

How was lipid A mixed with OVA? Was it an emulsion?   

Line 90:  "… mice were injected subcutaneously…"

Where were mice injected subcutaneously? In the back?

Line 100: "… anti-mouse IgG, IgG1, IgG2b, IgG3, and IgA… "

At what concentrations?

Line 100: "… anti-mouse IgG, IgG1, IgG2b, IgG3, and IgA… "

Instead of 'and', should this be 'or'?

Line 213: Discussion

Please remove the references to the figures within the discussion. This is distracting. The figures have already been reported within the Results section.

Author Response

Adjuvant activity of synthetic lipid A of Alcaligenes, a gut-associated lymphoid tissue-resident commensal bacterium, to augment antigen-specific IgG and Th17 responses in systemic vaccine.

Y.Wang et al.

This is a well constructed and executed study and a well-crafted paper. The authors have determined that lipid A from Alcaligenes is an effective adjuvant when using OVA as an antigen. It may be a well suited Th17-type adjuvant for systemic vaccination due to stimulation of dendritic cells.

Line 88:  "… 1 mg/mL and then mixed with OVA that was diluted in PBS for immunization." How was lipid A mixed with OVA? Was it an emulsion?  

No. It was just mixed without emulsion procedure. We have added this information at line 88 - 89 on page 2.

Line 90:  "… mice were injected subcutaneously…"

Where were mice injected subcutaneously? In the back?

Mice were injected subcutaneously in the back. We have added this information at line 91 on page 3

Line 100: "… anti-mouse IgG, IgG1, IgG2b, IgG3, and IgA… "

At what concentrations?

Thank you for the notice. The antibodies were used at the concentration of a dilution of 1:4,000 (antibodies: solution [v/v]). We have added this information at line 102 on page 3.

Line 100: "… anti-mouse IgG, IgG1, IgG2b, IgG3, and IgA… "

Instead of 'and', should this be 'or'?

Since we did not measure IgA, we have corrected them as… anti-mouse IgG, IgG1, IgG2b, and IgG3….

Line 213: Discussion

Please remove the references to the figures within the discussion. This is distracting. The figures have already been reported within the Results section.

Accordingly, we have corrected.

Thank you again for all of important comments to improve our study.

Round 2

Reviewer 1 Report

Reviewer questions were adequately answered by the authors

Author Response

Thank you for your cosideration.